# Governance and competitiveness evaluation of China's financial asset management corporations

**Guangcan Huang** [ORCID] *, **Jinqi Su**

School of Economics and Management, Xi'an University of Posts and Telecommunications, Yanta District, Xi'an City, Shaanxi Province, China

* hgc@xupt.edu.cn

## Abstract

With the development of non-performing assets market and the requirement of maintaining national financial security and stability, asset management corporations (AMC) have set up in recent years, facing the dilemma of governance reform and competitiveness improvement. This paper puts forward the Governance-Performance-Competitiveness theory of asset management corporations, and constructs the comprehensive competitiveness evaluation index system of based on internal and external governance mechanisms, and studies the factors and degrees of internal and external governance affecting competitiveness by combining principal component analysis method and grey correlation analysis method. The empirical results show that the asset scale and profitability of asset management corporations directly determine their competitiveness level to a large extent, and the correlation between external governance mechanism and competitiveness is stronger than that of internal governance mechanism. Clustering and grouping are conducted based on the size of competitiveness, and China's local asset management corporation system presents a diamond structure. China's local financial asset management corporations should strengthen the top-level design of internal governance, attach importance to external governance, expand and strengthen the asset scale, and carry out full-chains business while implementing a differentiated development model to achieve sustainable development when adhering to the principal business of non-performing assets.

**Data Availability Statement:** All relevant data are within the manuscript and its Supporting Information files. As a primary indicator, decision-making mechanism includes three secondary indicators: capital scale, market power, and innovation ability. The total assets and net assets

## 1. Introduction

Around 2012, the upward pressure on non-performing loans (NPLs) in the banking industry that has experienced the impact of the financial crisis has increased, and the scale of shadow banks continued to expand. A number of local financial asset management corporations began to establish in succession, with the aim of strengthening the acquisition and disposal of non-performing assets and maintaining the stability of the financial system in various regional spaces. This is a common practice of the international community to address the risk of non-performing loans [1]. From 2014 to the end of 2016, 42 local financial asset management

data are derived from the 2021 annual report or entity credit rating report of each financial asset management corporation, and the two data reflect the capital scale. As an important data to measure market power, the proportion of operating and disposal income of non-performing assets in main business income is derived from the entity credit rating reports of various financial asset management corporations in 2022. Innovation ability is mainly reflected in two aspects: funding expenditure and innovation activities, which are measured by research and development (R&D) expenditure and whether they have post-doctoral training qualifications. Research and development expenditure is derived from the 2021 consolidated profit statement of various financial asset management corporations. Whether the enterprise has post-doctoral training qualifications can be obtained through the recruitment information on the official website or the website of the China's Post-doctoral Management Committee. If the enterprise has a post-doctoral workstation or innovation practice base, 10 points will be obtained, otherwise 0 point will be obtained. The secondary indicators of the incentive mechanism are characterized by profitability, including three tertiary indicators, namely, rate of return on equity, operating revenue, and net profit, which are derived from the 2021 annual report and the main body credit rating report of various financial asset management corporations. The secondary indicators of the supervision mechanism are divided into risk control ability and financing ability, with current ratio and asset liability ratio selected respectively. The current ratio is the ratio of current assets to current liabilities, and the data is derived from the consolidated balance sheets in the 2021 annual reports of various financial asset management corporations. The asset liability ratio is the ratio of total assets to total liabilities, and the data is derived from the consolidated balance sheets in the 2021 annual reports of various financial asset management corporations. ESG mainly embodies social responsibility in financial asset management corporations, which is closely related to the state-owned nature and financial functions of financial asset management corporations. Social responsibility is first manifested in the most basic tax behavior, with tax data derived from the income tax in the consolidated profit statement of various financial asset management corporations in their 2021 annual reports. The second tertiary indicator is whether the enterprise has issued a social responsibility report. If it is issued, 10 points will be given, and if it is not issued, 0 point will be given. The result of an entity's credit rating is a reflection

corporations have begun to operate. In addition to the four national financial asset management corporations established in the 1990s, as of 2023, there have been more than 59 financial asset management corporations. The business model of non-performing assets is becoming increasingly mature, and market competitors are constantly pouring in. Various financial asset management corporations have begun to explore their own corporate governance models, and focus on improving the competitiveness of enterprises brought about by governance performance. Against the backdrop of huge global debt and the continuous explosion of local debt in China, the mission of China's financial asset management corporations has become even more challenging. The prerequisite for maintaining the stability of China's financial system is how to achieve its own governance transformation and possess international competitiveness. The main purpose of this study is to understand the level of competitiveness of China's financial asset management corporations and the impact of internal and external governance mechanisms on competitiveness, and to help China's financial asset management corporations find sustainable strategies to enhance their competitiveness.

The possible marginal contributions and innovations of this paper are as follows: Firstly, a theory of "governance-performance-competitiveness" for financial asset management corporations is proposed, in which improved governance leads to enhanced competitiveness; Secondly, a comprehensive evaluation index system for competitiveness based on internal and external governance mechanisms was constructed, and ESG practices were included in the evaluation; Third, using principal component analysis to measure the comprehensive competitiveness of financial asset management corporations, and using this as a reference sequence for grey correlation analysis to determine the extent of the impact of internal and external governance mechanisms on competitiveness.

## 2. Literature review

Corporate governance is an important issue that Chinese enterprises have to face in their transformation, upgrading, and pursuit of sustainable development. The issues of ownership structure and corporate governance have always been the focus of researches [2], and they are interdependent and evolving. China's corporate governance model presents a characteristic of being centered on the board of directors, which is based on the distribution of managers' matters and powers [3]. The separation of ownership, management power, and control power represents different stages of enterprise development, and enterprise also has different corporate governance priorities on different stages. The shift from the checks and balances of the new three boards of directors to the introduction of external directors and supervisors is a governance manifestation of the transformation of an enterprise from corporatization to collectivization [4]. With the continuous development of financial asset management corporations, the transformation of corporations into groups is the only way to enhance competitiveness. Grouping operations inevitably involve more stakeholders, and the social responsibility of financial asset management corporations has become a new and important issue in corporate governance. The establishment of a social responsibility committee within the board of directors is very common among the world's top 500 corporations in the United States and the United Kingdom [5]. The ultimate actual controller of the vast majority of financial asset management corporations is the local government in China. The attributes of state-owned enterprises are obvious. While actively assuming social responsibilities, there are inevitably some or other state-owned enterprise governance challenges, such as the absence of owners and the diversification of business objectives [6]. This requires that the governance of financial asset management corporations must coordinate internal and external interests and clarify the

of capital market opportunities. The data is derived from the 2022 entity credit rating report of various financial asset management corporations for 2021. The rating results are divided into three grades and nine levels. If the rating is AAA, 10 points will be obtained, 5 points will be obtained if AA, 0 points will be obtained if A, -5 points will be obtained if B, and -10 points will be obtained if C. The product market opportunity is measured by the scale of the non-performing loan balance in the banking in the region where it is located, and the data is derived from the China Regional Financial Operation Report (2022) or news information from the People's Bank of China. The balance of non-performing loans in the banking in the region where the four national financial asset management corporations are located is nationwide, while the data for local financial asset corporations is the balance of non-performing loans in the banking in the province where they are registered. The manager market mechanism affects the environment for financial asset management to tap talent. The manager market opportunity indicator is expressed as employee compensation payable. The more financial asset management corporations spend on this item, the more representative their ability to tap and absorb talent in the talent market. The data is derived from the consolidated balance sheets in the 2021 annual reports of various financial asset management corporations. The study selects 15 tertiary indicators to reflect the competitiveness differences caused by internal and external governance, and uses SPSS 26.0 to conduct descriptive statistics on them. The standard deviation and coefficient of variation are given. From the coefficient of variation, it can be seen that the selection of these 15 indicators is relatively reasonable and has a strong ability to identify competitiveness differences.

**Funding:** Guangcan Huang accepts funding from the fund project: Research Project on Major Theoretical and Realistic Issues in Philosophy and Social Sciences in Shaanxi Province: China Financial Asset Management Research (Grant number: 2022HZ1004). The funders had no role in study design, data collection and analysis, decision to publish, or preparation of the manuscript.

**Competing interests:** The authors have declared that no competing interests exist.

responsibilities of the shareholders' meeting, board of directors, board of supervisors, and senior managers, in particular, strengthen the supervision effect on the board of directors.

The corporate governance of banks will have a significant impact on their loan quality [7]. As the market entity for non-performing loan disposal, financial asset management corporations' own governance performance is also related to the quality of non-performing asset disposal. The results of corporate governance are expressed in the form of corporate governance performance, mainly reflected in financial data [8]. Whether the internal and external governance of a financial asset management corporation is effective determines the quality of presentation of asset size and profitability in financial statements. Currently, most of the research on governance performance evaluation focuses on listed corporations such as securities corporations and insurance corporations [9–11]. There is no research on the governance and competitiveness evaluation of local financial asset management corporations in China.

There are many applications of corporate governance performance evaluation methods. Zhang and Yin (2020) proposed three types of evaluation methods for corporate governance: profit evaluation method, value evaluation method, and balance evaluation method [12]. However, these three mainstream methods are all very single in measuring corporate governance performance, using only one financial indicator to represent corporate governance. Due to the fact that corporate governance performance is a complex indicator with multiple attributes [13], it is obviously not completed and scientific to measure it from one aspect. Chen (2018) comprehensively reflects governance performance from four indicators: corporation operating costs, corporation value, corporation revenue, and corporation compliance with regulations, but the relationship between these indicators and their relationship with governance is not well reflected [14]. Therefore, in the study of corporate governance performance evaluation, a comprehensive indicator evaluation system is usually constructed to systematically reflect governance performance. Li, Wang, and Li (2015) constructed an evaluation index system for corporate governance management entropy, breaking out of the constraints of internal control and financial indicators [15]. Tang and Zhang (2013) constructed a corporate governance indicator evaluation system based on the corporate governance structure and analyzed the factors affecting corporate governance using the generalized principal component analysis (GPCA) [16]. The evaluation of competitiveness is also the same [17]. Although the principal component model can effectively evaluate competitiveness, the internal and external governance mechanisms that affect competitiveness should be further explored as influencing factors.

## 3. Governance-performance-competitiveness theory of financial asset management corporations

Corporate governance is a systematic institutional arrangement for enterprises. As a modern enterprise, financial asset management corporations also have a complete set of modern governance models. A modern governance model can produce enterprise performance that conforms to the development trend of the times, which in turn manifests itself as a corporation's core competitiveness. The purpose of corporate governance is to coordinate the incentives and motivations of managers and shareholders, and to minimize the total cost of managers' self-interested behavior, thereby improving the quality of financial statements and shareholders' equity of the corporation. Financial asset management corporations are born to solve non-performing assets and maintain the stability of the financial system, with an inherent governance mission. Therefore, internal and external governance of corporations is crucial for the development and competitiveness improvement of financial asset management corporations.

### 3.1 General framework of corporate governance for financial asset management corporations

Corporate governance mainly includes two parts: governance structure and governance mechanism, which together constitute a corporation's governance model (Fig 1). The goal is to improve governance performance, directly reflected in the improvement of the corporation's competitiveness.

The governance structure is constructed based on the ownership structure, which determines the composition of the shareholders' general meeting. The shareholders' general meeting is the highest authority and decision-making institution of a financial asset management corporation, which authorizes and generates the board of directors, board of supervisors, and senior management. The ownership structure, shareholders' meeting, board of directors, board of supervisors, and senior managers together constitute the governance structure and directly affect the internal governance mechanism [18]. Internal governance mechanisms are an important component of governance mechanisms, including decision-making mechanisms, incentive mechanisms, supervision mechanisms, and ESG (Environmental, Social, and Corporate Governance) practices. The effective implementation of the decision-making mechanism, incentive mechanism, and supervision mechanism depends on whether the top-level design of the corporate governance structure is reasonable and advanced. ESG is a connecting point of internal and external governance [5]. Only if the external governance mechanism is smooth can ESG practice be reflected through the internal governance mechanism and profoundly affect the business development of financial asset management corporations. As another important component of governance mechanisms, external governance mechanisms are divided into capital market mechanisms, product market mechanisms, and manager market mechanisms. These three external governance mechanisms reflect the evaluation of financial asset management corporations in the capital market, their space in the product market, and their exploration in the manager market. External governance mechanisms also affect corporate governance performance and are reflected in corporate competitiveness. Research has shown that for banks, external governance such as creditor retention and talent development for senior management are more important than shareholder governance [19]. At present, financial asset management corporations are paying more attention to the construction of internal governance and the effective operation of external governance mechanisms, and are moving towards ESG practice.

### 3.2 Sources of competitiveness of financial asset management corporations

Based on the theory of enterprise innovation, starting from the perspectives of organizational innovation in corporate governance and business innovation in corporation upgrading, combined with market governance and government regulation, the enterprise knowledge system of financial asset management corporations is jointly formed (Fig 2), and this is used as a core competency and strategic resource to further comb the sources of enterprise competitiveness of financial asset management corporations [20], providing a theoretical basis for the establishment of competitiveness evaluation index systems.

Innovation is an important driving force for promoting the sustainable transformation and development of enterprises. Enterprise innovation behavior can be reflected in different levels such as system, management, and technology. Specifically, the innovation of financial asset management corporations can be divided into organizational innovation and business innovation. Organizational innovation generally refers to the adjustment and transformation of an enterprise's organizational structure and management methods as the external and internal conditions change. Its purpose is to enable the enterprise to adapt to new development trends

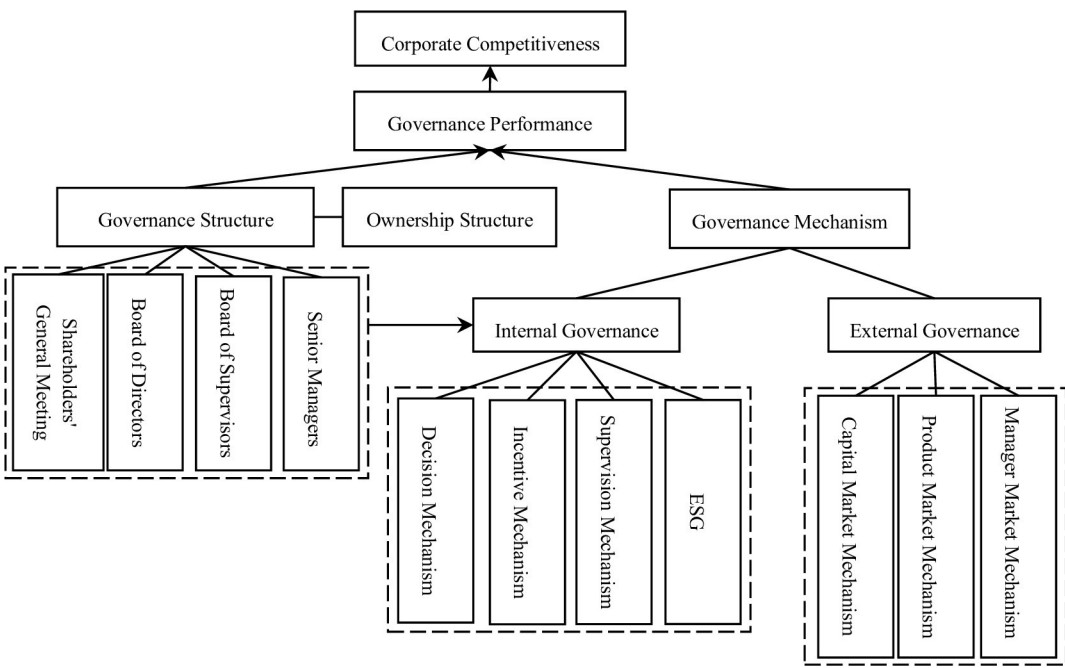

**Fig 1. Relationship between corporate governance and competitiveness of financial asset management corporations.**

and improve the efficiency of organizational activities. Business innovation generally refers to the continuous development of enterprises through establishing new business systems that adapt to the new environment and continuously exploring the market based on a profound understanding of the changes in the market and policy environment. Organizational innovation belongs to the category of corporate governance, while business innovation belongs to the category of corporate upgrading. The two complement and promote each other. The mission and role of financial asset management corporations are very special, coupled with China's strict supervision of the separate operation of the financial market, and their own business innovation space is very limited compared to organizational innovation. For the innovative behavior of financial asset management corporations, the task of corporate governance is more fundamental than the task of corporate upgrading.

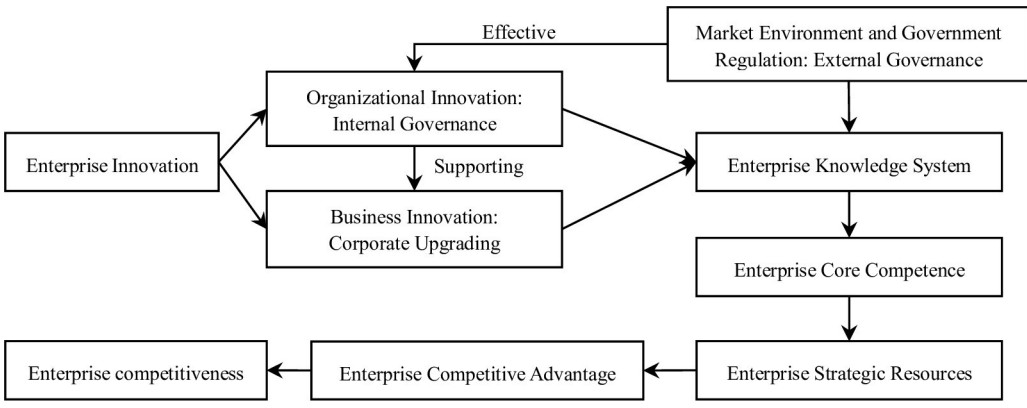

**Fig 2. Sources of competitiveness of financial asset management corporations.**

Corporate governance mechanisms are divided into internal governance and external governance, and together with corporate upgrading, constituting an enterprise's knowledge system. The internal governance of financial asset management corporations is based on the formal institutional arrangement of corporate governance recognized by law, while external governance is the institutional arrangement of corporate governance made by financial asset management corporations to adapt to the market and policy environment. The internal governance activities of a corporation are mainly reflected in the game state between shareholders, directors, supervisors, and managers, as well as the institutional design to encourage or restrain managers. As an effective supplement to internal governance, external governance activities of a corporation are mainly concentrated in external aspects of the corporation such as the product market, capital market, manager market, and regulatory policies [21]. Organizational innovation, business innovation, and external governance collectively constitute the knowledge system of an enterprise, and are the main sources of core competencies for financial asset management corporations.

## 4. Analysis on the competitiveness of financial asset management corporations

By the end of 2022, there were a total of 59 local AMC nationwide, two more than the end of 2020. Currently, the financial asset management corporation system in China's non-performing asset market has formed a 5+59 national-local pattern, and is still evolving. Many financial asset management corporations are deeply involved in China's non-performing asset market, and there are significant differences in the performance of corporate entities' internal and external governance mechanisms on financial data. Based on the governance-performance-competitiveness theory of financial asset management corporations, evaluating and analyzing the competitiveness of different financial asset management corporations is crucial to understanding the pattern of China's AMC.

### 4.1 Construction of competitiveness evaluation index system for financial asset management corporations

According to the theoretical model of governance-performance-competitiveness, the internal and external governance of financial asset management corporations will determine strategic resources and form enterprise competitiveness, which is expressed in the form of operational performance. Therefore, the competitiveness evaluation index system of financial asset management corporations is divided into two categories: internal governance and external governance, corresponding to decision-making mechanisms, incentive mechanisms, supervision mechanisms, ESG and capital market mechanisms, product market mechanisms, manager market mechanisms. These governance mechanisms can be quantified by selecting financial data from annual reports through segmented secondary indicators, forming 15 tertiary indicators (Table 1). These 15 tertiary indicators can reflect capital scale, market power, innovation ability, profitability, risk control, financing ability, social responsibility, entity credit rating, product market opportunities, and talent development ability.

The internal governance decision-making mechanism of financial asset management corporations is one of the most core mechanisms for modern enterprises to operate corporations through corporate governance structure. The efficient implementation of the decision-making mechanism will lead to a significant increase in the size of the entire enterprise, mainly reflected in the asset scale, market share, and innovation capacity building. Total assets and net assets can reflect the scale of capital. The proportion of main business in the management and disposal of non-performing assets is an important indicator of market power, which can reflect

**Table 1. Competitiveness evaluation index system of financial asset management corporations.**

| Large Category | Primary Indicators | Secondary Indicators | Tertiary Indicators |
|---|---|---|---|
| Internal Governance | Decision-Making Mechanism | Capital Scale | Total Assets |
| | | | Net Assets |
| | | Market Power | Main Business Structure of Operating and Disposing of Non-Performing Assets |
| | | Innovation Ability | R&D Expenditure |
| | | | Whether Qualified for Post Doctoral Training |
| | Incentive Mechanism | Profitability | Return on Equity |
| | | | Operating Revenue |
| | | | Net Profit |
| | Supervision Mechanism | Risk Control | Current Ratio |
| | | Financing Ability | Asset Liability Ratio |
| | ESG | Social Responsibility | Corporate Income Tax |
| | | | Whether to Publish Social Responsibility Report |
| External Governance | Capital Market Mechanism | Entity Credit Rating | Entity Credit Rating |
| | Product Market Mechanism | Product Market Opportunities | Market Size of Non-Performing Assets in the Region |
| | Manager Market Mechanism | Talent Development Ability | Employee Compensation Payable |

Note: Prepared by the author.

the position of the enterprise in the main business market. The R&D expenditure of an enterprise can largely reflect the investment in its innovation capacity, and some local financial asset management corporations have begun to build post-doctoral innovation and entrepreneurship practice bases, and have introduced post-doctoral researchers to serve the corporation's strategic planning department and board office to better play the role of the corporation's decision-making mechanism.

Incentive mechanism is an important mechanism for internal governance of financial asset management corporations. Whether this mechanism can be effectively implemented depends to a large extent on the profitability of the enterprise, mainly reflected in the return on equity, operating revenue, net profit, and other aspects. The rate of return on equity can standardized reflect the increase in asset income of a corporation, and the profitability of a corporation must also be based on the absolute scale of operating revenue and net profit. Financial asset management corporations operate on capital scale, so the combination of absolute scale and relative growth can more comprehensively reflect the profitability of financial asset management corporations. When financial asset management corporations have strong profitability, net profit and internal governance will make the incentive mechanism operate effectively.

The supervisory mechanism is an important mechanism for the supervisory board of financial asset management corporations to perform their duties. Financial asset management corporations mainly engage in the acquisition and disposal of non-performing assets, and the entire operational process itself is fraught with significant financial risks. This requires financial asset management corporations to pay more attention to the performance of the duties of the supervisory board than other enterprises. Research has shown that strengthening regulation and risk management can significantly reduce the scale of non-performing loans in the European banking [22]. A strict risk management and control system is of great significance for the operation of financial asset management corporations, which is also an important criterion for judging whether a financial asset management corporation is mature. Although the four major national financial asset management corporations have been listed for equity

financing, local financial asset management corporations are still unlisted enterprises, and their financing ability can reflect to the outside whether the enterprise supervision mechanism is effective. The board of supervisors ensures the effective operation of the supervision mechanism, with the structure and dynamic changes of corporate liabilities as the core focus, while risk control and financing ability are important indicators. The current ratio can reflect the short-term risk control level of an enterprise, and the level of financing ability depends on the asset liability ratio.

Environmental, social and Corporate Governance (ESG) assessment is a more inclusive corporate assessment that has also profoundly influenced investors' investment choices, business operations, and social values. Embedding social responsibility into business processes leads to innovation cycles and enhances corporate competitiveness [23]. China's companies with better performance in social responsibility have an advantage in bank loans [24]. The vast majority of financial asset management corporations are state-owned enterprises, directly controlled by state-owned capital or the government, and clearly bear a portion of social responsibility. Currently, financial asset management corporations have not disclosed environmental information in their respective audit reports. Some well-developed financial asset management corporations will choose to publish social responsibility reports, such as China Cinda Asset Management Co., Ltd. and Shanghai State-owned Asset Management Co., Ltd. The social responsibility report can comprehensively reflect the value concept of ESG evaluation actively practiced by financial asset management corporations, while in the financial statements, it is mainly reflected in the corporate income tax amount.

External and internal governance mechanisms jointly constitute the core competitiveness of financial asset management corporations. The external governance of financial asset management corporations is first manifested in the capital market mechanism. Of the 59 local financial asset management corporations, 28 have issued or still issue corporate bonds for the market, and third-party market rating agencies will conduct entity and bond credit ratings on 28 local financial asset management corporations. In the policy environment of limited IPO issuance, the main body credit ratings of local financial asset management corporations can reflect their own capital market opportunities. If the ratings are good, they have more space and ability to achieve in-depth business development, and also establish good capital market opportunities for possible future IPO issuance.

The product market opportunities of financial asset management corporations are mainly concentrated in the acquisition and disposal market of non-performing assets. The scale of non-performing assets determines the number of market opportunities for the main business of local financial asset management corporations. When a region has a large scale of non-performing assets, the value of the existence and operation of local financial asset management corporations in the region is greater. The larger the scale of the acquisition and disposal of non-performing assets, the more in-depth and smooth it is possible for financial asset management corporations that primarily engage in this business to achieve investment returns in related fields.

Local financial asset management corporations are quasi-financial institutions (China's Banking and Insurance Regulatory Commission had identified) that have emerged and established in recent years, with a unique financial position and specific business scope, which requires a large amount of specialized high-level human capital to support. Managers of financial asset management corporations have certain professional qualifications in legal affairs, finance, accounting, research, and business negotiations. The manager market mechanism is an important aspect of the external governance mechanism of financial asset management corporations. The manager market structure directly affects the talent source of financial asset management corporations, and thus affects a corporation's core competitiveness.

## 4.2 Data selection and source

As a primary indicator, decision-making mechanism includes three secondary indicators: capital scale, market power, and innovation ability. The total assets and net assets data are derived from the 2021 annual report or entity credit rating report of each financial asset management corporation, and the two data reflect the capital scale. As an important data to measure market power, the proportion of operating and disposal income of non-performing assets in main business income is derived from the entity credit rating reports of various financial asset management corporations in 2022. Innovation ability is mainly reflected in two aspects: funding expenditure and innovation activities, which are measured by research and development (R&D) expenditure and whether they have post-doctoral training qualifications. Research and development expenditure is derived from the 2021 consolidated profit statement of various financial asset management corporations. Whether the enterprise has post-doctoral training qualifications can be obtained through the recruitment information on the official website or the website of the China's Post-doctoral Management Committee. If the enterprise has a post-doctoral workstation or innovation practice base, 10 points will be obtained, otherwise 0 point will be obtained.

The secondary indicators of the incentive mechanism are characterized by profitability, including three tertiary indicators, namely, rate of return on equity, operating revenue, and net profit, which are derived from the 2021 annual report and the main body credit rating report of various financial asset management corporations.

The secondary indicators of the supervision mechanism are divided into risk control ability and financing ability, with current ratio and asset liability ratio selected respectively. The current ratio is the ratio of current assets to current liabilities, and the data is derived from the consolidated balance sheets in the 2021 annual reports of various financial asset management corporations. The asset liability ratio is the ratio of total assets to total liabilities, and the data is derived from the consolidated balance sheets in the 2021 annual reports of various financial asset management corporations.

ESG mainly embodies social responsibility in financial asset management corporations, which is closely related to the state-owned nature and financial functions of financial asset management corporations. Social responsibility is first manifested in the most basic tax behavior, with tax data derived from the income tax in the consolidated profit statement of various financial asset management corporations in their 2021 annual reports. The second tertiary indicator is whether the enterprise has issued a social responsibility report. If it is issued, 10 points will be given, and if it is not issued, 0 point will be given.

The result of an entity's credit rating is a reflection of capital market opportunities. The data is derived from the 2022 entity credit rating report of various financial asset management corporations for 2021. The rating results are divided into three grades and nine levels. If the rating is AAA, 10 points will be obtained, 5 points will be obtained if AA, 0 points will be obtained if A, -5 points will be obtained if B, and -10 points will be obtained if C.

The product market opportunity is measured by the scale of the non-performing loan balance in the banking in the region where it is located, and the data is derived from the *China Regional Financial Operation Report (2022)* or news information from the People's Bank of China. The balance of non-performing loans in the banking in the region where the four national financial asset management corporations are located is nationwide, while the data for local financial asset corporations is the balance of non-performing loans in the banking in the province where they are registered.

The manager market mechanism affects the environment for financial asset management to tap talent. The manager market opportunity indicator is expressed as employee

compensation payable. The more financial asset management corporations spend on this item, the more representative their ability to tap and absorb talent in the talent market. The data is derived from the consolidated balance sheets in the 2021 annual reports of various financial asset management corporations.

The study selects 15 tertiary indicators to reflect the competitiveness differences caused by internal and external governance, and uses SPSS 26.0 to conduct descriptive statistics on them (Table 2). The standard deviation and coefficient of variation are given. From the coefficient of variation, it can be seen that the selection of these 15 indicators is relatively reasonable and has a strong ability to identify competitiveness differences.

### 4.3 Empirical analysis

**4.3.1 Mathematical model of principal component and grey correlation analysis.** The study selected 22 financial asset management corporations with 2021 annual report (audit report) and 2022 entity credit rating from 5 national and 59 local financial asset management corporations to conduct a principal component analysis on competitiveness. From the original sample data, a matrix Y is obtained, expressed as:

$$Y = \begin{bmatrix} Y_{11} & Y_{12} & \cdots & Y_{1,15} \\ Y_{21} & Y_{22} & \cdots & Y_{2,15} \\ \cdots & \cdots & \cdots & \cdots \\ Y_{22,1} & Y_{22,2} & \cdots & Y_{22,15} \end{bmatrix} \tag{1}$$

$Y_{ij}$ is the tertiary indicator of item j of the financial asset management corporation i.

(1) Standardizing the original matrix. Due to the inconsistent dimensions of the elements in the matrix Y and the large gap among absolute quantities, Z-Score standardization is performed before principal component analysis in order to prevent excessively capturing the characteristics of certain indicator information. The standardization formula is as follows:

$$X_{ij} = \frac{Y_{ij} - \bar{Y}_j}{\delta_j} \tag{2}$$

$X_{ij}$ is the standardized data, $\bar{Y}_j$ is the average value of the tertiary index of item j, $\delta_j$ is the standard deviation of the tertiary index of item j.

(2) Calculating correlation coefficient matrix. Assume that $r_{ij}$ is the correlation coefficient between indicator $X_{ii}$ and indicator $X_{ij}$ after standardization. The correlation coefficient is expressed as:

$$r_{ij} = \frac{Cov\left(X_{ii}, X_{ij}\right)}{\delta_{ii} \cdot \delta_{ij}} \tag{3}$$

$Cov(X_{ii}, X_{ij})$ is a covariance between indicator $X_{ii}$ and indicator $X_{ij}$. $\delta_{ii}$ and $\delta_{ij}$ respectively are standard deviations of indicator $X_{ii}$ and indicator $X_{ij}$. Then, the correlation coefficient matrix $R = XX^T/(n-1)$ formed by the original sample data can be obtained. R is a positive

**Table 2. Descriptive statistics of competitiveness evaluation indicators for financial asset management corporations.**

| Evaluating Indicator | Average Value | Median | Standard Deviation | Variance | Coefficient of Variation |
|---|---|---|---|---|---|
| Total Assets | 2539.429 | 278.975 | 5067.372 | 25678263.173 | 1.995 |
| Net Assets | 340.193 | 97.880 | 540.792 | 292455.781 | 1.590 |
| Proportion of Non-Performing Asset Operation in Main Business | 1160.945 | 557.132 | 1745.199 | 3045718.982 | 1.503 |
| R&D Expenditure | 362.666 | 0.000 | 868.203 | 753776.948 | 2.394 |
| Whether Qualified for Post-Doctoral Training | 3.182 | 0.000 | 4.767 | 22.727 | 1.498 |
| Ratio of Return On Equity | 5.705 | 6.765 | 5.225 | 27.296 | 0.916 |
| Business Revenue | 163.915 | 19.435 | 349.432 | 122102.658 | 2.132 |
| Net Profit | 12.415 | 5.325 | 37.371 | 1396.625 | 3.010 |
| Current Ratio | 153.305 | 150.730 | 76.274 | 5817.725 | 0.498 |
| Asset Liability Ratio | 67.311 | 70.924 | 17.559 | 308.307 | 0.261 |
| Tax Expenses | 7.775 | 1.245 | 15.748 | 248.013 | 2.026 |
| Whether To Publish Social Responsibility Report | 1.364 | 0.000 | 3.513 | 12.338 | 2.576 |
| Entity Credit Rating | 8.636 | 10.000 | 2.279 | 5.195 | 0.264 |
| Scale Of Non-Performing Loans in the Local Banking | 5838.569 | 1226.590 | 10698.539 | 114458728.926 | 1.832 |
| Employee Compensation Payable | 8.621 | 0.840 | 19.190 | 368.244 | 2.226 |

Note: Statistical results are output from SPSS 26.0.

definite matrix, expressed as:

$$R = \begin{bmatrix} r_{11} & r_{12} & \cdots & r_{1,15} \\ r_{21} & r_{22} & \cdots & r_{2,15} \\ \cdots & \cdots & \cdots & \cdots \\ r_{22,1} & r_{22,2} & \cdots & r_{22,15} \end{bmatrix} \quad (4)$$

(3) Solving out the eigenvalues and eigenvectors of the covariance matrix. Calculating the 15 non-negative eigenvalues $\lambda_i = (\lambda_{1i}, \lambda_{2i}, \ldots, \lambda_{15,i})(i = 1, 2, \ldots, 15)$ of the characteristic equation $|R - \lambda I| = 0$ from the above correlation coefficient matrix R. The characteristic vector corresponding to the characteristic value is expressed as $\mu_i = (\mu_{1i}, \mu_{2i}, \ldots, \mu_{15,i})$. This constitutes a new combination of indicator variables:

$$F_m = \sum_{i=1}^{15} \mu_i \cdot X_i \quad (5)$$

Among them, m is the principal component number, and i is the tertiary index of item i.

(4) Calculating the contribution rate and cumulative contribution rate of the principal components. Calculating the variance percentage and cumulative contribution rate of each initial characteristic value respectively. The percentage of variance is also known as the information contribution rate, which can be expressed as:

$$b_i = \frac{\lambda_i}{\sum_{i=1}^{15} \lambda_i}, i = 1, 2, \ldots, 15 \quad (6)$$

$b_i$ is an information contribution rate of principal component $F_m$.

The cumulative contribution rate can be expressed as:

$$\alpha_i = \frac{\sum_{i=1}^{i} \lambda_i}{\sum_{i=1}^{15} \lambda_i}, i = 1, 2, \ldots, 15 \tag{7}$$

$\alpha_i$ is the cumulative information contribution rate of principal component $F_m$.

(5) Determining the equation expression of each principal component. From the results of the total variance interpretation, the study selects components with a characteristic root greater than 1 as the principal component to determine the number and cumulative contribution rate. Dividing the coefficient in the component matrix by the square root of each characteristic root to determine the functional relationship between each principal component and 15 original indicators. Based on this, calculating the scores and rankings of 22 financial asset management corporations in each principal component.

(6) Calculating the comprehensive competitiveness evaluation index of financial asset management corporations. The study uses the contribution rate of characteristic values to assign weights to each principal component, thereby constructing a comprehensive evaluation model for the competitiveness of financial asset management corporations, expressed as $Z = \sum_{i=1}^{m} b_i \cdot F_m$, then calculating the score index, and ranking it to assess the competitiveness.

Based on the results of the above principal component analysis, the comprehensive competitiveness evaluation index $Z_i$ of 22 financial asset management corporations can be obtained. In addition to the 15 tertiary indicators in the comprehensive evaluation index system of competitiveness that has been constructed, the grey correlation method is used to analyze the impact of 15 tertiary indicators on the comprehensive evaluation index.

(1) Determining reference sequences that reflect the competitiveness characteristics of financial asset management corporations and comparison sequences that affect their competitiveness characteristics. The reference sequence is recorded as $U_0'$, specifically expressed as $U_0'$ = $(U_0(1), U_0(2),\ldots,U_0(i)), i = 1, 2, \ldots, 22$, $i$ represents the number of financial asset management corporations participating in the evaluation. The comparison sequence is recorded as $U_1', U_2', \ldots, U_k', k = 1, 2, \ldots, 15$, $k$ represents the number of tertiary indicators in the comprehensive evaluation index system of competitiveness.

(2) Doing dimensionless processing on each original sequence. Since the dimensions of the 15 tertiary indicators are not consistent, considering eliminating the impact of different dimensions, the Z-Score standardization processing method described above is used to obtain a new reference sequence $U_0$ and a new comparison sequence $U_1, U_2, \ldots, U_k$.

(3) In order to reduce dimensions and better reflect the correlation between the impact of internal governance and external governance on competitiveness, comparison sequence $U_1, U_2,\ldots, U_k$ using the mean method to generate a new comparison sequence $W_1, W_2,\ldots, W_l$, $l = 1, 2, \ldots, 7$, based on each secondary indicator category, where $l$ represents the number of secondary indicators in the comprehensive evaluation index system of competitiveness.

(4) Calculating the absolute value of dimensionless processed data to form a new sequence:

$$\Delta_{(0,i)}(l) = |U_0(l) - W_l|, l = 1, 2, \ldots, 7 \tag{8}$$

(5) Finding the minimum and maximum difference in absolute value difference sequence $\Delta_{(0,i)}(l)$:

$$\min_{l}\left(\min_{i}\left(\Delta_{(0,i)}(l)\right)\right) = \min_{l}\left(\min_{i}|U_0(l) - W_l|\right) \tag{9}$$

$$\max_{l}\left(\max_{i}\left(\Delta_{(0,i)}(l)\right)\right) = \max_{l}\left(\max_{i}|U_0(l) - W_l|\right)$$

(6) Calculating the correlation coefficient between each secondary indicator of internal and external governance and the comprehensive evaluation index of competitiveness of financial asset management corporations:

$$\xi_l(i) = \frac{\min_{l}\left(\min_{i}|U_0(l) - W_l|\right) + \rho \cdot \max_{l}\left(\max_{i}|U_0(l) - W_l|\right)}{\Delta_{(0,i)}(l) + \rho \cdot \max_{l}\left(\max_{i}|U_0(l) - W_l|\right)} \tag{10}$$

Where, $\rho$ is the resolution coefficient, usually taken as 0.5, which is generally considered as $0 < \rho < 1$. The smaller $\rho$, the greater the difference in correlation coefficients, and the better the discrimination effect.

(7) Calculating the correlation degree between each secondary indicator of internal and external governance and the comprehensive evaluation index of competitiveness of financial asset management corporations:

$$\gamma = \frac{1}{N}\sum_{i=1}^{N}\xi_l(i), N = 1, 2, \ldots, 22 \tag{11}$$

**4.3.2 Empirical results of principal component analysis.** The study used SPSS 26.0 to first conduct KMO and Bartlett sphericity tests on standardized matrix elements (Table 3). The KMO test value is 0.657, so the original hypothesis can be rejected to a certain extent, and the selected data has a certain correlation. The results of Bartlett's sphericity test show that these raw data have strong significance, so the original assumption that the correlation coefficient matrix is a unit matrix is rejected. Overall, there is a certain correlation between the selected 15 tertiary indicators, and principal component analysis is appropriate.

Next, the study put the standardized sample data matrix into SPSS 26.0 for total variance analysis. From the Table 4, it can be seen that the initial characteristic roots corresponding to the first four principal components are greater than 1, and the cumulative contribution rate of the first four principal components reaches 85.7%. Therefore, the four principal components can largely reflect the information of the competitiveness indicator system of financial asset management corporations.

The component matrix is also calculated by SPSS 26.0. From the Table 5, it can be seen that some scale indicators in principal component F1 have higher factor loadings, while the profitability indicators in principal component F2 have higher factor loadings. Principal component F3 and principal component F4 respectively indicate that the factor loadings of the two indicators, namely, the entity credit rating and liquidity ratio, are higher. Therefore, the four principal components represent the scale dimension, profit dimension, capital market opportunity dimension, and risk control dimension, respectively. From the component matrix, it can be seen that the main factors affecting the competitiveness of financial asset management corporations are concentrated in the asset scale and operating volume.

**Table 3. KMO and Bartlett tests.**

| KMO Test | Bartlett Sphericity Test | | |
|---|---|---|---|
| | Approximate Chi-Square | Freedom | Significance |
| 0.657 | 566.114 | 105 | 0.000 |

Note: The test results are output by SPSS 26.0.

The initial characteristic roots of the component matrix coefficients and tertiary indicators were obtained through SPSS analysis. The coefficient of the principal component regression expression depends on the component matrix coefficient and the square root of the characteristic root. From this, four principal component regression expressions are calculated and sorted out:

$$F_1 = 0.339X_1 + 0.329X_2 - 0.137X_3 + 0.333X_4 + 0.246X_5 - 0.103X_6 + 0.333X_7 + 0.207X_8 - 0.005X_9 + 0.202X_{10} + 0.313X_{11} + 0.227X_{12} + 0.135X_{13} + 0.321X_{14} + 0.333X_{15} \quad (12)$$

$$F_2 = 0.027X_1 + 0.082X_2 + 0.359X_3 + 0.115X_4 - 0.272X_5 + 0.575X_6 + 0.115X_7 + 0.486X_8 - 0.036X_9 - 0.092X_{10} - 0.039X_{11} + 0.142X_{12} - 0.350X_{13} - 0.138X_{14} + 0.155X_{15} \quad (13)$$

$$F_3 = -0.048X_1 + 0.068X_2 - 0.510X_3 - 0.033X_4 - 0.120X_5 + 0.130X_6 - 0.020X_7 + 0.269X_8 + 0.416X_9 - 0.485X_{10} - 0.069X_{11} + 0.204X_{12} + 0.388X_{13} - 0.152X_{14} + 0.003X_{15} \quad (14)$$

$$F_4 = 0.100X_1 + 0.021X_2 + 0.237X_3 + 0.056X_4 - 0.195X_5 - 0.338X_6 + 0.043X_7 - 0.192X_8 + 0.658X_9 - 0.250X_{10} + 0.173X_{11} + 0.038X_{12} - 0.433X_{13} + 0.154X_{14} + 0.036X_{15} \quad (15)$$

**Table 4. Interpretation of total variance.**

| Component | Initial Characteristic Value | | | Extract the Sum of Squares of the Loadings | | |
|---|---|---|---|---|---|---|
| | Total | Percent Variance | Cumulative % | Total | Percent Variance | Cumulative % |
| 1 | 8.438 | 56.256 | 56.256 | 8.438 | 56.256 | 56.256 |
| 2 | 1.829 | 12.194 | 68.450 | 1.829 | 12.194 | 68.450 |
| 3 | 1.389 | 9.257 | 77.707 | 1.389 | 9.257 | 77.707 |
| 4 | 1.199 | 7.993 | 85.700 | 1.199 | 7.993 | 85.700 |
| 5 | 0.774 | 5.158 | 90.857 | | | |
| 6 | 0.445 | 2.968 | 93.825 | | | |
| 7 | 0.366 | 2.438 | 96.263 | | | |
| 8 | 0.298 | 1.985 | 98.248 | | | |
| 9 | 0.138 | 0.922 | 99.171 | | | |
| 10 | 0.076 | 0.510 | 99.681 | | | |
| 11 | 0.039 | 0.262 | 99.943 | | | |
| 12 | 0.007 | 0.048 | 99.991 | | | |
| 13 | 0.001 | 0.006 | 99.997 | | | |
| 14 | 0.000 | 0.003 | 100.000 | | | |
| 15 | 6.532E-05 | 0.000 | 100 | | | |

Note: The analysis results are output from SPSS 26.0.

**Table 5. Component matrix.**

| Evaluating Indicator | Component | | | |
|---|---|---|---|---|
| | **F1** | **F2** | **F3** | **F4** |
| Total Assets | **0.986** | 0.037 | -0.057 | 0.109 |
| Net Assets | **0.956** | 0.111 | 0.080 | 0.023 |
| Proportion of Non-Performing Asset Operation in Main Business | -0.398 | **0.485** | -0.601 | 0.260 |
| R&D Expenditure | **0.967** | 0.155 | -0.039 | 0.061 |
| Whether Qualified for Post-Doctoral Training | **0.715** | -0.368 | -0.141 | -0.213 |
| Ratio of Return on Equity | -0.298 | **0.778** | 0.153 | -0.37 |
| Business Revenue | **0.968** | 0.156 | -0.024 | 0.047 |
| Net Profit | 0.600 | **0.657** | 0.317 | -0.210 |
| Current Ratio | -0.015 | -0.049 | 0.490 | **0.720** |
| Asset Liability Ratio | **0.587** | -0.125 | -0.572 | -0.274 |
| Tax Expenses | **0.910** | -0.053 | -0.081 | 0.189 |
| Whether to Publish Social Responsibility Report | **0.659** | 0.192 | 0.241 | 0.042 |
| Entity Credit Rating | 0.391 | -0.473 | **0.457** | -0.474 |
| Scale of Non-Performing Loans in the Local Banking | **0.933** | -0.186 | -0.179 | 0.169 |
| Employee Compensation Payable | **0.966** | 0.210 | 0.004 | 0.039 |

Note: The analysis results are output from SPSS 26.0.

X1, X2, . . ., X15 in the above expression are all standardized values of the original values of 15 tertiary indicators. From the expressions of the above four principal components, the scores of 22 financial asset management corporations in the scale dimension (F1), profit dimension (F2), capital market opportunity dimension (F3), and risk control dimension (F4) can be calculated, as shown in the Table 6.

The principal component F1 contains most of the data information in the competitiveness evaluation indicator system, so its score and ranking are closer to the real situation. According to F1 scores, the top four financial asset management corporations are China Cinda Asset Management Co., Ltd., China Huarong Asset Management Co., Ltd., China Orient Asset Management Co., Ltd., and China Great Wall Asset Management Co., Ltd., all of which are the earliest four national financial asset management corporations. The four belong to the first echelons of China's financial asset management corporations in terms of asset scale, operating revenue, operating expenses, and social responsibility, which are not comparable to local financial asset management corporations. The top five local financial asset management corporations ranked according to the F1 component are Shanghai State-owned Asset Management Co., Ltd., Zhongyuan Asset Management Co., Ltd., China Resources YuKang Asset Management Co., Ltd., Shaanxi Financial Asset Management Co., Ltd., and Zhejiang Zheshang Asset Management Co., Ltd. These five financial asset management corporations are local leaders in terms of size and scale. This principal component F2 can reflect the profitability of each enterprise to a certain extent. Based on the financial data of 2021, China Cinda Asset Management Co., Ltd. still ranks first, while Xinjiang Financial Investment Asset Management Co., Ltd. and Xiamen Asset Management Co., Ltd. rank second and third. This principal component considers more the net profit generated by the main business rather than the asset size and enterprise size. The two principal components F3 and F4 contain relatively little data information, but in the constructed competitiveness evaluation indicator system, they still affect the entity credit rating and liquidity ratio, respectively. Just examining the ranking of F3 and F4 scores does not well reflect the reality, but they do have an impact on the competitiveness of each

**Table 6. Scores of principal components of some financial asset management corporations.**

| Financial Asset Management Corporation | Scale Volume F1 | | Profitability F2 | | Capital Market Opportunities F3 | | Risk Control F4 | |
|---|---|---|---|---|---|---|---|---|
| | score | rank | score | rank | score | rank | score | rank |
| China Cinda Asset Management | 8.13 | 1 | 2.01 | 1 | 0.76 | 6 | 0.16 | 8 |
| China Orient Asset Management | 5.72 | 3 | 0.81 | 5 | -0.27 | 13 | -0.62 | 16 |
| China Huarong Asset Management | 6.36 | 2 | -0.39 | 17 | -0.43 | 17 | 0.76 | 6 |
| China Great Wall Asset Management | 2.24 | 4 | -4.45 | 22 | -1.89 | 20 | 1.3 | 5 |
| Zhongyuan Asset | -0.39 | 6 | -1.64 | 21 | -0.01 | 12 | -0.81 | 17 |
| Shaanxi Financial Asset Management | -0.64 | 8 | 0.05 | 11 | 0.45 | 8 | -1.55 | 20 |
| Inner Mongolia Financial Asset Management | -1.57 | 17 | -0.81 | 19 | 2.63 | 1 | 1.94 | 1 |
| Shanghai State-owned Assets Management | -0.37 | 5 | 0.29 | 10 | 1.76 | 2 | -0.57 | 15 |
| Jiangsu Asset Management | -1.41 | 15 | -0.05 | 14 | 1.22 | 3 | -0.21 | 11 |
| Suzhou Asset Management | -1.49 | 16 | 0.72 | 7 | -0.95 | 19 | -0.25 | 14 |
| Zheshang Asset Management, Zhejiang | -0.81 | 9 | -0.11 | 15 | 0.86 | 4 | -0.21 | 12 |
| Goho Asset Management | -1.71 | 19 | 0.72 | 6 | 0.11 | 10 | 1.45 | 3 |
| Xiamen Asset Management | -1.85 | 20 | 1.4 | 3 | -1.93 | 21 | -0.21 | 13 |
| China Industry Asset Management | -1.25 | 12 | 0.02 | 13 | -0.35 | 15 | -0.17 | 10 |
| Jiangxi Financial Asset Management | -1.64 | 18 | 0.87 | 4 | -0.85 | 18 | 0.68 | 7 |
| Shandong Financial Asset Management | -0.83 | 10 | -0.66 | 18 | 0.81 | 5 | 0.16 | 9 |
| Henan Asset Management | -1.25 | 13 | -0.3 | 16 | 0.04 | 11 | -1.14 | 19 |
| Chasing Asset Management, Hu'nan | -1.39 | 14 | 0.48 | 9 | 0.22 | 9 | -0.84 | 18 |
| Guangzhou Asset Management | -1.05 | 11 | 0.02 | 12 | -0.38 | 16 | -1.6 | 22 |
| China Resources YuKang Asset Management | -0.61 | 7 | -1.38 | 20 | -0.28 | 14 | -1.56 | 21 |
| Gansu Asset Management | -1.94 | 21 | 0.51 | 8 | 0.76 | 7 | 1.93 | 2 |
| Xinjiang Financial Investment Asset Management | -2.24 | 22 | 1.9 | 2 | -2.3 | 22 | 1.37 | 4 |

enterprise. Therefore, the analysis of comprehensive evaluation scores and their rankings will be necessary and more meaningful.

The study constructs a comprehensive evaluation model for the competitiveness of financial asset management corporations that includes principal components, with the contribution rates of the four principal components as weights:

$$Z = 0.563F_1 + 0.122F_2 + 0.093F_3 + 0.080F_4 \tag{16}$$

The principal component score table has calculated the four principal component values of 22 financial asset management corporations, and substituted them into the above formula to obtain a comprehensive evaluation index of the competitiveness of 22 financial asset management corporations (Table 7). Due to the participation of the four national financial asset management corporations in the evaluation, their huge capital scale and profitability have led to negative comprehensive competitiveness evaluation indexes for other local asset management corporations, but this does not affect their ranking order.

China Cinda Asset Management Co., Ltd., China Huarong Asset Management Co., Ltd., and China Orient Asset Management Co., Ltd., with total assets of over trillion yuan and strong business capabilities, strongly occupy the top three nationwide. Next is China Great Wall Asset Management Co., Ltd. Although the total assets and net assets of China Great Wall Asset Management Co., Ltd. are only half of the top three, and the net profit loss in 2021, its competitiveness is still stronger than any other local financial asset management corporation.

The top three competitiveness comprehensive evaluation indexes of local financial asset management corporations are Shanghai State-owned Asset Management Co., Ltd., Zhejiang Zheshang Asset Management Co., Ltd., and Shaanxi Financial Asset Management Co., Ltd.

**Table 7. Comprehensive evaluation index and ranking of competitiveness of 22 financial asset management corporations in 2021.**

| Financial Asset Management Corporations | Comprehensive Evaluation Index of Competitiveness | Ranking |
|---|---|---|
| China Cinda Asset Management | 4.90589 | 1 |
| China Orient Asset Management | 3.24447 | 3 |
| China Huarong Asset Management | 3.55391 | 2 |
| China Great Wall Asset Management | 0.64645 | 4 |
| Zhongyuan Asset | -0.48538 | 9 |
| Shaanxi Financial Asset Management | -0.43637 | 7 |
| Inner Mongolia Financial Asset Management | -0.58294 | 10 |
| Shanghai State-owned Assets Management | -0.05485 | 5 |
| Jiangsu Asset Management | -0.70327 | 12 |
| Suzhou Asset Management | -0.85938 | 20 |
| Zheshang Asset Management, Zhejiang | -0.40627 | 6 |
| Goho Asset Management | -0.74866 | 14 |
| Xiamen Asset Management | -1.06704 | 21 |
| China Industry Asset Management | -0.74746 | 13 |
| Jiangxi Financial Asset Management | -0.84183 | 19 |
| Shandong Financial Asset Management | -0.45968 | 8 |
| Henan Asset Management | -0.82783 | 18 |
| Chasing Asset Management, Hu'nan | -0.77075 | 16 |
| Guangzhou Asset Management | -0.75205 | 15 |
| China Resources YuKang Asset Management | -0.66263 | 11 |
| Gansu Asset Management | -0.80492 | 17 |
| Xinjiang Financial Investment Asset Management | -1.13362 | 22 |

Note: The results were calculated by the author based on a comprehensive evaluation model for the competitiveness of financial asset management corporations.

Shanghai State-owned Assets Management Co., Ltd. ranks first among local financial asset management corporations in terms of total assets, net assets, and net profits. Its size, scale, and profitability are excellent, but its main business of acquisition and disposal of non-performing assets accounts for only 9.98%, with the vast majority of its revenue derived from other equity operations and financial investments. This is mainly due to the huge initial funds of Shanghai State-owned Assets Management Co., Ltd., rich local financial asset management resources, and mature governance model. The shareholders of Shanghai State-owned Assets Management Co., Ltd. are scattered, with a high degree of marketization. The corporate governance structure makes Shanghai State-owned Assets Management Co., Ltd. have a greater degree of operational autonomy, and have a state-owned asset background. They possess the resources of the Shanghai government and actively assume social responsibilities. This enables enterprises to have better governance performance [25]. Shanghai State-owned Assets Management Co., Ltd. regularly publishes annual reports and social responsibility reports on its official website. It is also the only local financial asset management corporation to publish social responsibility reports, fully reflecting the excellent development strength of local financial asset management corporations. Zhejiang Zheshang Asset Management Co., Ltd. follows closely in various important financial data. Although Shaanxi Financial Asset Management Co., Ltd. is not particularly large in terms of capital scale and volume, it ranks first among all local financial asset management corporations in terms of ratio of return on equity , and has strong

profitability. However, when investigating the contribution of its main business of non-performing assets, it is not satisfactory. Shaanxi Financial Asset Management Co., Ltd. and Shanghai State-owned Asset Management Co., Ltd. are similar in that their considerable net profits are not derived from the contribution of the main business of non-performing assets, but from equity investment revenue. Although Shandong Financial Asset Management Co., Ltd. and Zhongyuan Asset Management Co., Ltd. have a large capital scale and volume, their ratio of return on equity in 2021 was only 2.15% and 1.42%, far below the average of 6.92% for all local financial asset management corporations. This is an important reason why the comprehensive competitiveness evaluation rankings of the two companies in 2021 fell behind.

**4.3.3 Empirical results of grey correlation analysis.** The study uses the comprehensive competitiveness evaluation index of 22 financial asset management corporations in 2021 as a reference sequence, and uses 7 primary indicators that reflect their internal and external governance as a comparison sequence, and performs Z-Score standardization on the sequence values to address the impact of different dimensions, then generate a standardized sequence matrix of internal and external governance and competitiveness of financial asset management corporations (Table 8).

The study conducts a grey correlation measurement on the standardized sequence matrix of internal and external governance and competitiveness of the above-mentioned financial asset management corporations to determine the impact of each level of indicators on competitiveness (Table 9).

First, the impact of external governance mechanisms of financial asset management corporations is greater than that of internal governance mechanisms. Firstly, among the external governance mechanisms of financial asset management corporations, the product market mechanism and the manager market mechanism have the greatest correlation with competitiveness, with 0.960 and 0.944, respectively, ranking the top two among the 7 primary indicators. The size of the non-performing asset market in the region fundamentally affects the revenue scale of a financial asset management corporation, and the ability to tap professionals in the field of non-performing assets can affect the internal governance of the entire corporation at the human capital level, which is called talent supports development.

Second, the degree of ESG influence in the internal governance mechanism of financial asset management corporations is greater than that of decision-making, supervision, and incentive mechanisms. ESG is an organic combination of environmental, social, and corporate governance, a concentrated manifestation of maintaining the stability of the financial system and assuming social responsibility. It is a key point to connect internal and external governance, and an important reference that financial asset management corporations must consider when developing their businesses. The corporate social responsibility of American listed companies will have a positive impact on corporate value and improve business performance [26]. Financial asset management corporations with strong competitiveness and a large volume insist on publishing social responsibility reports to the community, including the four national financial asset management corporations and Shanghai State-owned Asset Management Co., Ltd. However, local financial asset management corporations have not yet taken a substantive step in ESG, and ESG practice will be the most important sign of leading enterprises becoming bigger and stronger.

Third, the correlation of decision-making mechanisms, supervision mechanisms, and incentive mechanisms in internal governance mechanisms has gradually decreased. Decision-making mechanism is one of the most important mechanisms in internal governance mechanisms. Its effective operation determines the scale and volume of financial statements, and these indicators directly determine the competitiveness of an enterprise. The supervision mechanism in financial and internal control effectively compensates for internal decision-

**Table 8. Standardized sequence matrix of internal and external governance and competitiveness of financial asset management corporations.**

| | Decision Mechanism | Incentive Mechanism | Supervision Mechanism | ESG | Capital Market Mechanism | Product Market Mechanism | Manager Market Mechanism | Comprehensive Evaluation Index of Competitiveness |
|---|---|---|---|---|---|---|---|---|
| China Cinda Asset Management | 1.796 | 5.771 | 0.697 | 2.693 | 0.598 | 2.071 | 3.176 | 2.972 |
| China Orient Asset Management | 1.504 | 4.460 | 0.377 | -0.010 | 0.598 | 2.071 | 2.268 | 1.966 |
| China Huarong Asset Management | 1.382 | 1.698 | 0.860 | 2.599 | 0.598 | 2.071 | 1.665 | 2.153 |
| China Great Wall Asset Management | 0.472 | -5.876 | 0.632 | 0.459 | 0.598 | 2.071 | 0.167 | 0.392 |
| Zhongyuan Asset | -0.074 | -1.477 | 0.135 | -0.435 | 0.598 | -0.420 | -0.410 | -0.294 |
| Shaanxi Financial Asset Management | -0.048 | 1.308 | 0.075 | -0.361 | 0.598 | -0.491 | -0.371 | -0.265 |
| Inner Mongolia Financial Asset Management | -0.547 | -1.274 | 0.352 | -0.418 | 0.598 | -0.485 | -0.434 | -0.353 |
| Shanghai State-owned Assets Management | -0.326 | -0.255 | -1.271 | 0.995 | 0.598 | -0.474 | -0.397 | -0.033 |
| Jiangsu Asset Management | -0.465 | -0.091 | -0.730 | -0.368 | 0.598 | -0.421 | -0.421 | -0.426 |
| Suzhou Asset Management | -0.413 | -0.152 | -0.472 | -0.372 | -1.595 | -0.421 | -0.420 | -0.521 |
| Zheshang Asset Management, Zhejiang | -0.488 | 0.372 | 0.776 | -0.296 | 0.598 | -0.431 | -0.281 | -0.246 |
| Goho Asset Management | -0.382 | -0.246 | 0.465 | -0.395 | -1.595 | -0.495 | -0.433 | -0.454 |
| Xiamen Asset Management | -0.120 | -0.060 | -0.619 | -0.422 | -1.595 | -0.486 | -0.376 | -0.647 |
| China Industry Asset Management | -0.249 | -0.417 | 0.656 | -0.421 | 0.598 | -0.486 | -0.434 | -0.453 |
| Jiangxi Financial Asset Management | -0.303 | -0.213 | 0.288 | -0.408 | -1.595 | -0.495 | -0.403 | -0.510 |
| Shandong Financial Asset Management | -0.285 | -1.153 | -0.036 | -0.387 | 0.598 | -0.431 | -0.392 | -0.279 |
| Henan Asset Management | -0.413 | -0.710 | -1.020 | -0.415 | 0.598 | -0.420 | -0.370 | -0.502 |
| Chasing Asset Management, Hu'nan | -0.382 | 0.654 | 0.087 | -0.413 | 0.598 | -0.482 | -0.415 | -0.467 |
| Guangzhou Asset Management | -0.418 | 0.085 | -0.393 | -0.353 | 0.598 | -0.422 | -0.408 | -0.456 |
| China Resources YuKang Asset Management | -0.129 | -1.155 | -0.423 | -0.428 | 0.598 | -0.499 | -0.422 | -0.402 |
| Gansu Asset Management | -0.405 | -0.801 | -0.272 | -0.412 | -1.595 | -0.410 | -0.445 | -0.488 |
| Xinjiang Financial Investment Asset Management | 0.293 | -0.469 | -0.167 | -0.432 | -1.595 | -0.515 | -0.447 | -0.687 |

Note: The sequence matrix is composed of the reference sequence and the comparison sequence after Z-Score standardization processing.

making and incentive actions, and is also crucial for improving governance and competitiveness.

**Table 9. Correlation and ranking of between competitiveness of financial asset management corporations and governance mechanisms.**

| | Internal Governance | | | | External Governance | | |
|---|---|---|---|---|---|---|---|
| | Decision-Making Mechanism | Incentive Mechanism | Supervision Mechanism | ESG | Capital Market Mechanism | Product Market Mechanism | Manager Market Mechanism |
| Correlation | 0.923 | 0.804 | 0.844 | 0.939 | 0.756 | 0.944 | 0.960 |
| Ranking | 4 | 6 | 5 | 3 | 7 | 2 | 1 |

**4.3.4 Systematic cluster analysis of comprehensive competitiveness evaluation index.**
Based on the above principal component analysis, the comprehensive competitiveness evaluation index of 22 financial asset management corporations in 2021 was calculated and ranked accordingly. In order to further conduct qualitative analysis of financial asset management corporations, the study needs to conduct a systematic cluster analysis of the comprehensive competitiveness evaluation index of each corporation in 2021.

Systematic clustering analysis uses the inter-group connection method as a clustering method to measure the square Euclidean distance of the interval to ensure that the aggregation result minimizes its average distance, and can well measure the comprehensive competitiveness evaluation index of each financial asset management corporation. Due to the fact that the four national financial asset management corporations are far superior to local financial asset management corporations in various financial indicators, and the gap in their comprehensive evaluation index is also very huge (-1.134~4.906), a unified clustering of 22 financial asset management corporations will result in a significant dichotomy between national and local financial asset management corporations. Although this is realistic and obvious, it cannot advance in-depth research and is of little significance. Therefore, it is necessary to cluster and analyze the competitiveness comprehensive evaluation indexes of four national financial asset management corporations and 18 local financial asset management corporations respectively.

Firstly, the comprehensive competitiveness evaluation indexes of four national financial asset management corporations are systematically clustered. According to the pedigree chart, the four national financial asset management corporations are divided into three echelons: the first echelon is China Cinda Asset Management Co., Ltd., the second echelon is China Huarong Asset Management Co., Ltd. and China Eastern Asset Management Co., Ltd., and the third echelon is China Great Wall Asset Management Co., Ltd. The competitiveness of these three echelons decreases in turn. As China Huarong Asset Management Co., Ltd. has a slightly stronger asset size and amount of tax payment than China Orient Asset Management Co., Ltd., but its liabilities and shareholders' equity have relatively deteriorated. The two corporations are more similar in terms of competitiveness and are classified as the second echelon.

In view of the significant differences in the competitiveness of national and local financial asset management corporations, in order to cluster local financial asset management corporations more effectively, a separate systematic clustering is hereby conducted. According to the pedigree chart, 18 local financial asset management corporations can be roughly divided into four echelons (Table 10). The leading enterprise of local financial asset management corporations is Shanghai State-owned Asset Management Co., Ltd., the first echelon corporation. This corporation has the highest shareholder equity in the region, and has a large capital scale and volume. It is also the only local financial asset management corporation that issues social responsibility reports. This shows that its internal and external governance performance is excellent. Although Shandong Financial Asset Management Co., Ltd. has total assets of over 100 billion yuan, its return on equity in 2021 was very low, at only 2.15%, and its competitiveness significantly weakened in that year. Shaanxi Financial Asset Management Co., Ltd., on the other hand, has performed well with a good ratio of return on equity, coupled with the

**Table 10. Cluster group of competitiveness of local financial asset management corporations.**

| Cluster Group | Name of Local Financial Asset Management Corporation | Competitiveness Level | Position Percentage |
|---|---|---|---|
| First Echelon | Shanghai State-owned Assets Management Co., Ltd | High | 5.6% |
| Second Echelon | Shandong Financial Asset Management., Zhejiang Zheshang Asset Management, Shaanxi Financial Asset Management, Zhongyuan Asset Management, Inner Mongolia Financial Asset Management | Mid-High | 27.8% |
| Third Echelon | Goho Financial Asset Management, China Industry Asset Management, Guangzhou Asset Management, Hunan Chasing Asset Management, Jiangsu Asset Management, China Resources YuKang Asset Management, Jiangxi Financial Asset Management, He'nan Asset Management, Suzhou Asset Management, Gansu Asset Management | Medium | 55.5% |
| Fourth Echelon | Xiamen Asset Management, Xinjiang Financial Investment Asset Management | Low | 11.1% |

Note: The author compiles it based on the comprehensive competitiveness evaluation index and the system cluster pedigree chart.

increase in size and market-oriented reform of the equity structure, and has a relatively high competitiveness among many local financial asset management corporations. More than 55.5% of local financial asset management corporations are concentrated in the third echelon, widely distributed in various regions, and various competitiveness evaluation indicators are well regulated, with a certain degree of competitiveness. However, Xiamen Asset Management Co., Ltd. and Xinjiang Financial Investment Asset Management Co., Ltd. are ranked in the fourth echelon, with relatively weak asset size, shareholder equity, and profitability, which is far from most local financial asset management corporations, and even the entity credit rating in the audit report is only grade AA.

Currently, the system of local financial asset management corporations in China presents a diamond-shaped structure. The leading role of local head financial asset management corporations is not obvious. Most corporations are located in the middle of the diamond structure, with serious homogenization of internal and external governance models, and unclear internal and external governance mechanisms that drive competitiveness improvement.

## 5. Conclusion and proposal

The governance models of local financial asset management corporations and national financial asset management corporations have many similarities, but there are also some differences, which widen the competitive gap between them. Although financial asset management corporations have a competitive relationship with banking related financial asset investment corporations and foreign alternative investment institutions in the non-performing asset market, their advantages and disadvantages are also relatively obvious.

### 5.1 Main conclusions

The competitiveness of China's financial asset management corporations is determined by their governance performance, mainly reflected in four dimensions: scale volume, profitability, capital market opportunities, and risk control. The internal and external governance mechanisms of enterprises affect the performance of governance. This paper uses principal component analysis to calculate and rank the competitiveness level of 22 financial asset management corporations in China. Study finds that financial asset management corporations with diversified main businesses, larger scale, more market-oriented equity structures, and stronger profitability have stronger competitiveness. According to the analysis of the grey correlation model, the impact of external governance mechanisms of financial asset management corporations is

greater than that of internal governance mechanisms. The impact of ESG in internal governance mechanisms of financial asset management corporations is greater than that of decision-making, supervision, and incentive mechanisms. The correlation degree of decision-making, supervision, and incentive mechanisms in internal governance mechanisms gradually decreases. The result of system clustering analysis indicates that the system of local financial asset management corporations in China exhibits a diamond structure, and majority of corporations is in the middle of the diamond structure, with severe homogenization of internal and external governance models.

## 5.2 Strategies for improving the competitiveness of local financial asset management corporations

Based on the above study results, the following strategies are proposed for improving the competitiveness of local financial asset management corporations.

**5.2.1 Emphasizing the top-level design of corporate governance structure to effectively promote the operation of internal governance mechanisms.** The corporate governance structure of financial asset management corporations and the establishment of their committees should be clear. The board of directors sets up a strategic development committee, a nomination and compensation committee, a risk management committee, an audit committee, and a connected transaction committee, which are responsible for the effective implementation of decision-making mechanisms, incentive mechanisms, and internal control mechanisms [18, 27]. The board of supervisors sets up a due diligence supervision committee and an internal control supervision committee, which respectively give full play to the functions of the supervision mechanism for the effective implementation of the incentive mechanism and internal control mechanism, and coordinate stakeholders within the framework of the shareholders' general meeting, to enable the operation of financial asset management corporations in a more stable state. The senior management is under the unified leadership of the board of directors, with specific executive functional departments based on the scope of work of each committee, to do a good job in the daily operation of the corporation. The committee of the board of directors and the supervisory committee of the board of supervisors are established by the group's staff cadres in accordance with the governance procedures, and their functions do not conflict with the functional departments under the senior management.

**5.2.2 Attaching importance to the role of external governance mechanisms in improving governance performance and enhancing competitiveness, and actively exploring ESG practices with the characteristics of local financial asset management corporations.** The correlation between the external governance mechanism of financial asset management corporations and the comprehensive competitiveness evaluation index is stronger than that of the internal governance mechanism, which is manifested in the region's non-performing asset market space, the ability to tap talent, and the evaluation of third-party rating agencies. The improvement of these capabilities or the improvement of the environment is significant for enhancing competitiveness. Local financial asset management corporations should integrate ESG practices into their business development and actively contribute to the disposal of non-performing assets and equity investment in areas such as green and low-carbon finance, common prosperity, industrial transformation and upgrading, and strategic emerging technology empowerment. In addition, local financial asset management corporations should issue social responsibility reports as a result or purpose of optimizing their internal and external governance models, and shoulder social responsibilities such as local environmental improvement, job creation, financial volunteer activities, and industrial development assistance while focusing on the main business of non-performing assets. Only when environmental, social, and

corporate governance information is sufficiently publicly disclosed can a local financial asset management corporation be considered to be actively practicing ESG and effective. This will profoundly affect the revenue scale and profitability of local financial asset management corporations.

**5.2.3 Increasing asset scale and profitability.** From the results of principal component analysis, asset size and corporation size directly and decisively affect the competitiveness of a financial asset management corporation, followed by profitability, both of which are equally important. Local financial asset management corporations are increasing their registered capital based on their own business conditions, and introducing strategic investors with government background, international management experience, or market resources as shareholders in due course. Under strict risk control and financial audit requirements, local financial asset management corporations appropriately enhance short-term debt capacity and the ability to handle short-term debt, and activate funds. The ultimate business goal is to achieve the growth of shareholders' equity at a significant growth rate, which requires achieving a high return on equity and expanding the scale of operating revenue under a certain asset size. The results of the return on equity should be fully considered in the development of all businesses.

**5.2.4 Actively exploring the core business layout of "non-performing assets main business, alternative investment banks, diversified comprehensive financial services", and creating the entire chain of financial asset management.** The business focus should always be on the acquisition and disposal of non-performing assets as the main business, and on the basis of its main business, it should carry out businesses such as equity investment, distressed bond investment, high-yield bond investment, fund management, financial leasing, supply chain finance, financial asset trading, and financial management, to achieve diversified comprehensive financial services around the main business of non-performing assets. From the perspective of Shanghai State-owned Assets Management Co., Ltd. and Shaanxi Financial Assets Management Co., Ltd., the revenue of the main business of non-performing assets is not the source of high ratio of return on equity, but is contributed by investment revenue. However, the resulting risks are large and the annual gap is also large. This problem is reflected in the 2021 financial statements of Zhongyuan Asset Management Co., Ltd. The market for the acquisition and disposal of non-performing assets has a significant periodicity, which depends on the cycle of financial risk management. If the main business of non-performing assets is in the recession, local financial asset management corporations have to develop diversified and comprehensive financial services around the main business to maintain the sustainability and stability of their own operations.

**5.2.5 Promoting a differentiated development model with local characteristics in the field of financial asset management.** There are significant differences between local financial asset management corporations and national financial asset management corporations, banking financial asset investment corporations, and foreign alternative investment institutions in terms of size, positioning, business, and advantages. Promoting a differentiated development model with regional economic development characteristics is the key to their transformation. Local financial asset management corporations should compete with national financial asset management corporations in their respective provinces, and serve the solution of non-performing assets in the characteristic and pillar industries of their respective regions. The financial asset investment corporations with banks in the field of debt-equity swap implement differentiated development, and more resources have been put into the area of non-performing debt-equity swap to avoid disorderly and low return competition with mature channels and parent bank resources of the banking system. In the face of competition from foreign alternative investment institutions, China's financial asset investment corporations should seize the disadvantage of foreign counterparts' lack of localization experience and

ability, give full play to local government resources, occupy more market share of non-performing assets, and cooperate with them in the international non-performing assets market to accelerate the pace of international exploration.

## Supporting information

**S1 Data.**
(XLSX)

## Acknowledgments

I am very grateful to the external reviewers and journal editors for their useful evaluations and suggestions on my research paper. There are no academic misconduct or interest disputes in this article. Guangcan Huang accepts funding from the fund project: Research Project on Major Theoretical and Realistic Issues in Philosophy and Social Sciences in Shaanxi Province: China Financial Asset Management Research (Grant number: 2022HZ1004). The funders had no role in study design, data collection and analysis, decision to publish, or preparation of the manuscript.

## Author Contributions

**Conceptualization:** Guangcan Huang.

**Data curation:** Guangcan Huang.

**Formal analysis:** Guangcan Huang.

**Funding acquisition:** Guangcan Huang.

**Methodology:** Guangcan Huang.

**Project administration:** Guangcan Huang.

**Resources:** Guangcan Huang, Jinqi Su.

**Software:** Guangcan Huang.

**Supervision:** Jinqi Su.

**Writing – original draft:** Guangcan Huang.

**Writing – review & editing:** Jinqi Su.

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
