## [Decision Letter · Decision Letter 0]

20 Apr 2023

PONE-D-23-09947Governance and Competitiveness Evaluation of China's Financial Asset Management CorporationsPLOS ONE

Dear Dr. Huang,

Thank you for submitting your manuscript to PLOS ONE. After careful consideration, we feel that it has merit but does not fully meet PLOS ONE’s publication criteria as it currently stands. Therefore, we invite you to submit a revised version of the manuscript that addresses the points raised during the review process.

We look forward to receiving your revised manuscript.

Kind regards,

JOANNA ROSAK-SZYROCKA, Assistant Professor

Academic Editor

PLOS ONE

"I accept funding from the fund project: Research Project on Major Theoretical and Realistic Issues in Philosophy and Social Sciences in Shaanxi Province: China Financial Asset Management Research (2022HZ1004). I am very grateful to the journal editors and anonymous reviewers for helpful comments on earlier drafts."

Please remove any funding-related text from the manuscript and let us know how you would like to update your Funding Statement. Currently, your Funding 

Statement reads as follows: 

"Guangcan Huang accepts funding from the fund project: Research Project on Major Theoretical and Realistic Issues in Philosophy and Social Sciences in Shaanxi Province: China Financial Asset Management Research (Grant number: 2022HZ1004).

Upon re-submitting your revised manuscript, please upload your study’s minimal underlying data set as either Supporting Information files or to a stable, public repository and include the relevant URLs, DOIs, or accession numbers within your revised cover letter. For a list of acceptable repositories, please see http://journals.plos.org/plosone/s/data-availability#loc-recommended-repositories. 

Any potentially identifying patient information must be fully anonymized.

Important: If there are ethical or legal restrictions to sharing your data publicly, please explain these restrictions in detail. Please see our guidelines for more information on what we consider unacceptable restrictions to publicly sharing data: http://journals.plos.org/plosone/s/data-availability#loc-unacceptable-data-access-restrictions.

Note that it is not acceptable for the authors to be the sole named individuals responsible for ensuring data access.

4. Please ensure that you include a title page within your main document. You should list all authors and all affiliations as per our author instructions and clearly indicate the corresponding author.

Reviewers' comments:

Reviewer's Responses to Questions

**Comments to the Author**

1. Is the manuscript technically sound, and do the data support the conclusions?

Reviewer #1: No

Reviewer #2: Yes

2. Has the statistical analysis been performed appropriately and rigorously? 

Reviewer #1: No

Reviewer #2: Yes

3. Have the authors made all data underlying the findings in their manuscript fully available?

Reviewer #1: No

Reviewer #2: Yes

4. Is the manuscript presented in an intelligible fashion and written in standard English?

Reviewer #1: No

Reviewer #2: Yes

5. Review Comments to the Author

Reviewer #1: Comments for authors:

1. The English language needs more work. There are many grammatical and typo mistakes in this manuscript. The paper needs to be edited by a native English speaker.

2. I suggest the authors revise the introduction of the study per the comments raised. The authors can also use the following points below as a guideline to help them come out with an interesting introduction that is more scientific.

Background & Significance: (What general background does the reader need in order to understand the manuscript and how important is it in the context of scientific research).

Problem definition: (What are the research questions to fill in the gaps of the existing knowledge body or methodology (Methods are not a contribution, but a tool to assess whether your hypothesis or predictions are supported or not supported)? I would like to see well developed arguments for predicting or proposing specific relationships in this study.

Motivations & Objectives: (Why are you conducting the study and what are the goals to achieve?)

3. I would like to suggest that authors should update the introduction, and results part. Specifically, the latest research trends, and in order to highlight the academic frontier of the research, the references of the recent year need to be referenced.

4. What is the methodological contribution of this paper? The author needs to pinpoint the exact methodological contribution of this present paper.

5. The author(s) need to compare their results (Each Findings) with past studies (what was provided in the article is not compares of results but an explanation of views from past authors) and in comparing the result from the empirical investigations the author(s) should as much as possible provide a recast of the comparison made and the supposed implications or advantages of the new finding made with those discovered by past authors. This will ensure justice to the extant literature and also evincing the superiority of the current findings over the past findings.

6. In the conclusions, in addition to summarizing the actions taken and results, please strengthen the explanation of their significance.

Reviewer #2: Dear Authors,

I found your study very interesting. There are some suggestions however, for improving your work:

1. Please add few lines about the methodology into the Abstract

2. Please explain how is your sample significant for your study by supporting this with previous studies

3. Please discuss your findings about indicators relevance and impact comparing to previous studies

4. All subsections from section 4 must be merged into Conclusions and practical recommendations section

6. PLOS authors have the option to publish the peer review history of their article (what does this mean?). If published, this will include your full peer review and any attached files.

Reviewer #1: No

Reviewer #2: No

<quillbot-extension-portal></quillbot-extension-portal>

---

## [Author Response · Author response to Decision Letter 0]

3 Jun 2023

Response to Reviewers

Dear external experts,

I am very grateful to PLoSONE for giving me the valuable opportunity to revise and improve my paper. I am also very grateful to the two experts for your helpful suggestions for my research. I have made improvements based on expert opinions and responded to this.

Reviewer # 1 believes that

1. There are some spelling errors in the initial draft of the paper. I carefully checked the paper and made careful spelling revisions. The meaning of "three boards and one layer" in Chinese is corporate governance structure, specifically referring to shareholders' meetings, board of directors, board of supervisors, and senior managers. I hereby replace it to eliminate the writing mistakes in Chinglish.

2. Revising the introduction based on expert opinions. The first paragraph of the introduction explains the research background and significance, the practical problems to be solved, and the goals to be achieved.

3. Updating references to highlight frontier and advanced academic researches. Such as: 

Tarchouna A., Jarraya B., Bouri A. Do Board Characteristics and Ownership Structure Matter for Bank Non-Performing Loans? Empirical Evidence from US Commercial Banks[J]. Journal of Management and Governance, 2022, 26: 479–518. https://doi.org/10.1007/s10997-020-09558-2

[24] Huang G., Ye F., Li Y. et al. Corporate Social Responsibility and Bank Credit Loans: Exploring the Moderating Effect of the Institutional Environment in China[J]. Asia Pacific Journal of Management, 2023, 40: 707–742. https://doi.org/10.1007/s10490-021-09800-x

Bouslah K., Hmaittane A., Kryzanowski, L. et al. CSR Structures: Evidence, Drivers, and Firm Value Implications[J]. Journal of Business Ethics, 2023, 185: 115–145. https://doi.org/10.1007/s10551-022-05219-6

Hopt K.J. Corporate Governance of Banks and Financial Institutions: Economic Theory, Supervisory Practice, Evidence and Policy [J]. European Business Organization Law Review, 2021, 22: 13–37. https://doi.org/10.1007/s40804-020-00201-z

4. The methodological contribution is clearly stated in the last part of the introduction. Based on internal and external governance mechanisms, a comprehensive evaluation index for enterprise competitiveness was constructed, and the factors and their degrees that affect the formation of competitiveness were studied by combining principal component analysis and grey relational analysis. This is a very innovative methodology.

5. Reviewers believe that a comparison should be made with past research to demonstrate the superiority and value of current research. The measurement of competitiveness is mostly based on countries or industries. I take China’s Financial Asset Management Corporation as the research object and construct a competitiveness evaluation index system based on corporate governance mechanism to measure China's AMC competitiveness in non-performing asset disposal market.

6. In the conclusion and Proposal, I added a discussion on the importance of measures and suggestions to highlight the theoretical and practical value of the research.

Reviewer # 2 believes that

1. Adding methodological content to the abstract of the paper. This paper proposes the theory of governance-performance-competitiveness for financial asset management corporations, and constructs a comprehensive competitiveness evaluation index system for financial asset management corporations based on internal and external governance mechanisms, and uses principal component analysis to measure and rank the competitiveness level of China’s financial asset management corporations, and establishes a grey correlation model to analyze the degree of correlation between competitiveness level and governance mechanisms. In addition, the paper explores the main layout of the non-performing asset disposal market through clustering and grouping.

2. The research sample for my paper is suitable and rich. There are currently 5 national financial asset management corporations and 59 local financial asset management corporations in China, but only 22 financial asset management corporations still publicly issue bonds in the market in 2021. According to Chinese financial law, if a financial enterprise publicly issues market bonds, it must disclose its main credit rating report or financial annual report to the society. Generally speaking, China’s corporations that are able to publicly disclose financial data are corporations with standardized operations, a certain scale, and status in their industry sector, which can largely reflect the true situation of this sector. Regarding the availability of financial data, I can only collect financial data from these 22 financial asset management corporations for research purposes. Regarding the selection of data for the 2021 financial annual report, this is because China's largest real estate company, Evergrande Group (www.evergrande.com), announced in 2021 that it was unable to fulfill its debt guarantee obligations, involving a debt of 2 trillion Chinese yuan. Local financial asset management corporations in China will therefore have good development opportunities, and each company is researching sustainable development strategies to enhance competitiveness. Therefore, selecting the financial data of 22 financial asset management corporations for 2021 as a sample is reasonable and appropriate, and can support the current research on financial asset management corporate governance in China.

3. Previous studies did not consider the dimensions of ESG (environmental, social, and governance) when constructing evaluation indicators for enterprise competitiveness. However, my study included ESG in the evaluation indicator system, and the ESG indicator used whether financial asset management corporations issued social responsibility reports as a proxy variable. From the operational practice of financial asset management corporations in China, the selection of this proxy variable is very appropriate, as only enterprises with more advanced asset management concepts will release social responsibility reports, which reflect the governance performance of financial asset management corporations.

4. Experts believe that the content of last part should be divided into two small parts: conclusions and measures. I added the main conclusion in the last section.

My research is very interesting and of great significance for the sustainable development of China’s financial asset management corporations and the understanding of the disposal entities in the China’s non-performing asset market. I am looking forward to my paper being published in PLoSONE. There are no disputes of interest or any academic misconduct in this paper. Please continue processing my manuscript with the editor. Thanks.

Regards,

Guangcan Huang

---

## [Decision Letter · Decision Letter 1]

4 Sep 2023

Governance and Competitiveness Evaluation of China's Financial Asset Management Corporations

PONE-D-23-09947R1

Dear Dr. Guangcan Huang

We’re pleased to inform you that your manuscript has been judged scientifically suitable for publication and will be formally accepted for publication once it meets all outstanding technical requirements.

Kind regards,

JOANNA ROSAK-SZYROCKA, Assistant Professor

Academic Editor

PLOS ONE

Additional Editor Comments (optional):

Please note that one reviewer found the way you responded to their comments difficult to assess. Having looked at their comments and you responses we feel that you have addressed them and done so clearly. In particular we note that the reviewer states you have not replied point by point, however it is our view that you have done so. The second reviewer has also assessed and confirmed that you have answered all the comments. 

Reviewers' comments:

Reviewer's Responses to Questions

**Comments to the Author**

1. If the authors have adequately addressed your comments raised in a previous round of review and you feel that this manuscript is now acceptable for publication, you may indicate that here to bypass the “Comments to the Author” section, enter your conflict of interest statement in the “Confidential to Editor” section, and submit your "Accept" recommendation.

Reviewer #1: (No Response)

Reviewer #2: All comments have been addressed

2. Is the manuscript technically sound, and do the data support the conclusions?

Reviewer #1: No

Reviewer #2: Yes

3. Has the statistical analysis been performed appropriately and rigorously? 

Reviewer #1: No

Reviewer #2: Yes

4. Have the authors made all data underlying the findings in their manuscript fully available?

Reviewer #1: No

Reviewer #2: Yes

5. Is the manuscript presented in an intelligible fashion and written in standard English?

Reviewer #1: No

Reviewer #2: Yes

6. Review Comments to the Author

Reviewer #1: The authors have not made the response of reviewers in well manner. The authors should reply one by one of each my points. Therefore, I can't take my decision on this version.

Reviewer #2: No further suggestions. The authors addressed all comments. I believe the paper can be published in its current form.

7. PLOS authors have the option to publish the peer review history of their article (what does this mean?). If published, this will include your full peer review and any attached files.

Reviewer #1: No

Reviewer #2: No

<quillbot-extension-portal></quillbot-extension-portal>

---

## [Editor Report · Acceptance letter]

26 Sep 2023

PONE-D-23-09947R1 

Governance and competitiveness evaluation of China's financial asset management corporations 

Dear Dr. Huang:

I'm pleased to inform you that your manuscript has been deemed suitable for publication in PLOS ONE. Congratulations! Your manuscript is now with our production department. 

Kind regards, 

on behalf of

Dr Carla Pegoraro 

Staff Editor

PLOS ONE